# Oxidized Phospholipids Regulate Tenocyte Function via Induction of Amphiregulin in Dendritic Cells

**DOI:** 10.3390/ijms25147600

**Published:** 2024-07-11

**Authors:** Veronica Pinnarò, Stefanie Kirchberger, Sarojinidevi Künig, Sara Gil Cantero, Maria Camilla Ciardulli, Giovanna Della Porta, Stephan Blüml, Adelheid Elbe-Bürger, Valery Bochkov, Johannes Stöckl

**Affiliations:** 1Center for Pathophysiology, Infectiology and Immunology, Institute of Immunology, Medical University of Vienna, 1090 Vienna, Austria; veronica.pinnaro@meduniwien.ac.at (V.P.); sarojinidevi.kuenig@meduniwien.ac.at (S.K.); sara.gilcantero@meduniwien.ac.at (S.G.C.); 2St. Anna Children’s Cancer Research Institute (CCRI), 1090 Vienna, Austria; stefanie.kirchberger@ccri.at; 3Department of Medicine, Surgery and Dentistry, University of Salerno, Via S. Allende, 84081 Baronissi, Italy; mciardulli@unisa.it (M.C.C.); gdellaporta@unisa.it (G.D.P.); 4Division of Rheumatology, Department of Internal Medicine III, Medical University of Vienna, 1090 Vienna, Austria; stephan.blueml@meduniwien.ac.at; 5Department of Dermatology, Medical University of Vienna, 1090 Vienna, Austria; adelheid.elbe-buerger@meduniwien.ac.at; 6Department of Pharmaceutical Chemistry, Institute of Pharmaceutical Sciences, University of Graz, 8010 Graz, Austria; valery.bochkov@uni-graz.at

**Keywords:** dendritic cells, tenocytes, fibroblasts, phospholipids oxidation, tendinopathies

## Abstract

Inflammation is a driving force of tendinopathy. The oxidation of phospholipids by free radicals is a consequence of inflammatory reactions and is an important indicator of tissue damage. Here, we have studied the impact of oxidized phospholipids (OxPAPC) on the function of human tenocytes. We observed that treatment with OxPAPC did not alter the morphology, growth and capacity to produce collagen in healthy or diseased tenocytes. However, since OxPAPC is a known modulator of the function of immune cells, we analyzed whether OxPAPC-treated immune cells might influence the fate of tenocytes. Co-culture of tenocytes with immature, monocyte-derived dendritic cells treated with OxPAPC (Ox-DCs) was found to enhance the proliferation of tenocytes, particularly those from diseased tendons. Using transcriptional profiling of Ox-DCs, we identified amphiregulin (AREG), a ligand for EGFR, as a possible mediator of this proliferation enhancing effect, which we could confirm using recombinant AREG. Of note, diseased tenocytes were found to express higher levels of EGFR compared to tenocytes isolated from healthy donors and show a stronger proliferative response upon co-culture with Ox-DCs, as well as AREG treatment. In summary, we identify an AREG-EGFR axis as a mediator of a DC-tenocyte crosstalk, leading to increased tenocyte proliferation and possibly tendon regeneration.

## 1. Introduction

Inflammation is a pivotal defense mechanism of our immune system [1]. Not only is it essential to protect us against invading pathogens, but also important against non-infectious diseases, such as injuries, and is also the starting point of tissue repair processes [2,3].

Tendinopathies are characterized by chronic inflammation that often leads to pain, swelling, and reduced tendon function [4,5,6]. This persistent inflammation results in ongoing discomfort and impaired healing, posing significant challenges in clinical treatment [7]. Understanding the inflammatory pathways, especially the dysregulation of cellular processes and the biochemical environment within the tendon matrix, is essential for developing targeted therapies to alleviate symptoms and promote recovery in tendinopathy patients [8,9].

The production and release of free reactive oxygen species (ROS) is a key component of inflammatory reactions [10,11]. A consequence of this process is the oxidation of phospholipids by ROS [12]. The modification of phospholipids by ROS causes dramatic chemical restructuring of oxidized phospholipids (OxPLs), such as oxidized 1-palmitoyl-2-arachidonoyl-sn-glycero-3-phosphocholine (OxPAPC) [13,14]. Such modified OxPLs are now recognized as potential danger signals, or alarmins, by a set of specific receptors including Toll-like receptors (TLRs) or scavenger receptors, interactions that can trigger a variety of novel biological activities [15]. The role of OxPLs has been studied in different inflammatory diseases and can have detrimental effects, such as arteriosclerosis, but also beneficial effects, such as in sepsis [14,16]. OxPLs are increasingly recognized as playing a role in a variety of normal and pathological states. OxPLs are implicated in the regulation of inflammation, angiogenesis, endothelial barrier function, immune tolerance, and other important processes [13]. For example, the treatment of endothelial cells with OxPLs typically leads to the induction of inflammatory molecules, such as IL-8 [17]. In contrast, treatment of human dendritic cells (DCs) with OxPAPC (Ox-DCs) has been shown to inhibit their potent T-cell stimulatory capacity, and to reduce the production of inflammatory cytokines, including IL-12, by epigenetic regulation [15,18,19].

Interestingly, recent studies demonstrated that OxPLs have significant influence on the biology of vascular smooth muscle cells (VSMCs), including proliferation, migration, differentiation, and production of extracellular matrix [20]. OxPLs were shown to stimulate the proliferation of VSMCs in vitro, and after the application of the lipids to the carotid artery, in vivo. Furthermore, in vitro and in vivo OxPL stimulation of VSMCs induced the expression of several extracellular matrix proteins, including the alpha 1 chain of type VIII collagen [21].

Taken together, the available data allow us to hypothesize that OxPLs may play a role in the pathogenesis of vascular smooth muscle cell-related diseases and illustrates the context-dependent mode of action of OxPLs, which, depending on the biological milieu, can induce different, even functionally opposite effects.

The potential role of OxPLs on the proliferation and collagen production of tenocytes has not been reported thus far. In view of the impact of OxPLs on VSMCs, in this study, we analyzed the functional consequences of OxPAPC treatment, a prototypic OxPL, on human tenocytes.

## 2. Results

### 2.1. Treatment with OxPAPC Does Not Alter Proliferation or Collagen Production in Human Tenocytes

Oxidized phospholipids (OxPLs) such as OxPAPC are known to have pleiotropic effects on cells. In order to study the functional role of OxPLs on tenocytes, we cultured primary human tenocytes obtained from the tendon of healthy donors (hTCs) and from diseased explants (dTCs) in the presence or absence of OxPAPC. Tenocytes, or specifically hTCs and dTCs, refer to the isolated human tendon progenitor stem cells (TPSCs) which have undergone at least two passages in culture.

First, we compared the proliferation and collagen production ability of the tenocytes in comparison to primary human fibroblasts (FBs) isolated from skin. The results presented in Figure 1 demonstrate that both types of tenocytes showed a lower proliferative capacity compared to FBs. Similarly, tenocytes produced lower amounts of type I collagen compared to FBs (Figure 1b). This functional difference was particularly obvious when comparing dTCs and FBs. However, although not statistically significant, the capability to grow and produce collagen was lower in dTCs than in hTCs, indicating a reduced functionality in dTCs.

Addition of OxPAPC to dTCs, hTCs or FBs had no visible impact on the morphology or viability of the cells (Figure 2). In order to analyze potential effects of OxPAPC in more detail, we performed proliferation and collagen production assays. The results shown in Figure 2 demonstrate that OxPAPC treatment of dTCs, hTCs or FBs did not affect the proliferation (Figure 2a) nor viability (Appendix A) of the cells or modulate their capacity to produce collagen (Figure 2b).

Thus, although OxPAPC is able to interact with cells via a variety of receptors or receptor-independent signaling mechanism, OxPAPC is not a direct activator or inhibitor of tenocytes or FBs.

### 2.2. OxPAPC-Treated Dendritic Cells (Ox-DCs) Stimulate the Proliferation of dTCs, hTCs, and FBs 

The interplay of tenocytes with immune cells, in particular macrophages, is recognized as an important factor in the regulation of tendon repair [5,22,23,24]. Recent studies have demonstrated the presence of resident dendritic cells (DCs) in tendons from man and animals. OxPAPC is a potent modulator of the function of DCs [15], so we wondered whether treatment of DCs might affect their ability to interact with and stimulate human tenocytes.

The results presented in Figure 3 demonstrate that untreated immature DCs (iDCs) showed a trend to increase the proliferation of the tested cells, but this effect was not significant. Ox-DCs were indeed able to enhance the proliferation of co-cultured dTCs and hTCs. However, the impact of Ox-DCs was particularly strong on dTCs, which proliferated as well as hTCs or FBs in the presence of Ox-DCs (Figure 3a). In contrast to TC, the proliferation of FBs induced by Ox-DCs was not increased compared to iDCs.

In order to analyze whether the support by Ox-DCs for hTCs, dTCs, and FBs depends on the direct contact between the cells or via soluble factors, we next tested the effect of supernatants (SNs) from Ox-DCs or iDCs on our cells. As shown in Figure 3b, the SN from Ox-DCs enhanced the proliferation of dTCs and FBs but not hTCs. In contrast to the proliferative response, neither Ox-DCs (Figure 4a) nor the SN of Ox-DCs (Figure 4b) altered collagen production by hTCs, dTCs or FBs. Thus, Ox-DCs seemingly promote the proliferation of dTCs and FBs primarily via the release of a soluble factor. In the case of hTCs, cell–cell contact is also involved in the induction of the response.

### 2.3. OxPAPC-Treated Dendritic Cells (Ox-DCs) Produce and Release Amphiregulin (AREG)

Stimulation of DCs with OxPAPC is accompanied with a dramatic functional reprogramming of the cells [15]. In order to find out which soluble factor released by Ox-DCs promotes the growth of dTCs and FBs, we performed microarray analyses of RNA from Ox-DCs and untreated DCs. Out of the 733 genes which were upregulated, amphiregulin (AREG) was one of the most prominent (51-fold increase) (Appendix A). AREG attracted our interest since it is a ligand for the epidermal growth factor receptor (EGFR) and is a non-classical product of DCs [25].

To confirm the microarray data, we first analyzed AREG mRNA expression in Ox-DCs by real-time RT-PCR. Results presented in Figure 5 demonstrate that the stimulation of immature DCs with OxPAPC induces the expression of AREG in a dose-dependent manner. Next, we analyzed whether AREG was detectable in the supernatant of Ox-DC cultures. We found that AREG was released in the supernatant of DC cultures upon stimulation with OxPAPC in a concentration-dependent manner (Figure 5b).

### 2.4. AREG Is a Growth Factor for dTCs and FBs

AREG is a well-known growth factor for different cell types and plays an important role in wound healing and tissue repair [26,27]. To investigate the impact of AREG on tenocytes or fibroblasts, we stimulated hTCs, dTCs, and FBs with increasing amounts of recombinant AREG (Figure 6). We observed that dTCs and FBs responded to AREG stimulation, and proliferation was significantly increased above 0.1 µg/mL of added AREG. In contrast, AREG did not induce proliferation of hTCs (Figure 6a).

In parallel, we tested the potential consequences of AREG stimulation of dTCs, hTCs, and FBs on their capacity to produce collagen. The results presented in Figure 6b show that in spite of the ability of AREG to stimulate the proliferation of dTCs and FBs, the production of collagen was not enhanced. While AREG had no influence on collagen production by the tenocytes used in this study, AREG stimulation of FBs inhibited collagen production when used at the highest dose of 1 µg/mL (Figure 6b).

### 2.5. dTCs Expresses EGFR

Human fibroblasts have been described to express the EGFR and respond to AREG (Figure 6) [28]. Whether tenocytes express the EGFR has not been reported. The finding that dTCs, but not hTCs, reacted to the supernatant of Ox-DCs or recombinant AREG raised the question of whether dTCs and hTCs express different levels of EGFR on their surface. Cell surface staining and subsequent analysis of EGFR expression by flow cytometry revealed that most of the dTCs express high amounts of EGFR, whereas only a few hTCs express the EGFR at very low levels (Figure 7).

Thus, differential expression levels of EGFR on dTCs and hTCs may be responsible for the differential responsiveness to AREG.

## 3. Discussion

OxPLs are endogenous alarmins that are formed during inflammation by reactive oxygen species [13,29]. The functional impact of OxPLs has been investigated in different disease models and on different cell types. Based on these studies, it is well established that OxPLs can initiate and modulate inflammatory reactions in a context-dependent mode of action [30,31]. Depending on the biological environment, OxPLs can induce different and even functionally opposite effects in cells. The role of OxPLs on human tenocytes has yet to be investigated. In view of the stimulatory potential of OxPLs on the proliferation, migration, differentiation, and production of extracellular matrix by vascular smooth muscle cells, we wondered whether OxPLs could have similar effects on tenocytes. Herein, we have analyzed the impact of the classical oxidized phospholipid OxPAPC on the function of human tenocytes and, for comparison, on human fibroblasts from the skin. We observed that treatment with OxPAPC did not alter the morphology, growth, or capacity to produce collagen by healthy or diseased tenocytes. Since OxPAPC is a known modulator of the function of dendritic cells, we then analyzed whether OxPAPC-treated immune cells might influence the fate of tenocytes. Treatment of immature, monocyte-derived dendritic cells with OxPAPC (Ox-DCs) was found to enhance the proliferation of tenocytes, particularly those from diseased tendon, via release of the growth factor amphiregulin (AREG), a ligand for EGFR [32,33]. Thus, inflammatory reactions in the tendon might lead to the oxidation of phospholipids which can then trigger the production AREG in resident DCs and thereby stimulate the proliferation of tenocytes.

DCs are part of the myeloid cell system and are known to be the most potent antigen-presenting cells of the immune system, pivotal for the induction of T-cell activation and proliferation [34,35]. The importance of the myeloid cells in tendon repair is well established [36,37]. In particular, macrophages are known to be essential for tendon regeneration [38,39,40,41,42,43]. The presence of DCs in the tendons has only recently been discovered, and their functional role in tendons remains unknown. Nonetheless, DCs have been reported to contribute to tissue repair and wound healing in several tissues [44,45,46,47,48,49]. Studies of OxPL effects on DCs demonstrate an overall reduction in T-cell stimulation capacity, resulting in strongly diminished T-cell proliferation [50]. OxPLs are also able to interfere with DC activation induced by TLR4 and TLR3 ligands. Pretreatment of DCs with OxPAPC inhibited the LPS-induced expression of costimulatory molecules CD40, CD80, and CD86, the surface expression of MHC-classes I and II, and the secretion of IL-12 and TNF [18]. Interestingly, the inhibition of IL-12 secretion by OxPAPC was mediated by attenuation of histone H3 phosphorylation, suggesting that OxPL action on DCs at least partly depends on epigenetic mechanisms [18,51]. These findings indicate that OxPLs have a deep impact on the functional repertoire of DC, shifting their mode of action from the induction and stimulation of adaptive immune responses towards tissue repair mechanisms, which might be at least in part induced via the release of amphiregulin (AREG).

Amphiregulin is a multifunctional factor important in wound healing and tissue repair, and a ligand for the EGFR [27,52]. Here, we found that the addition of recombinant AREG was indeed a significant trigger for tenocyte proliferation, but not collagen production. AREG is produced by different cell types [27,52,53]. It has previously been reported that DCs produce AREG upon stimulation with exogenous ATP [25]. However, in the study of Bles et al., the production of AREG by DCs had an adverse effect and led to an increase in tumor proliferation [25]. AREG was particularly effective on dTCs, but not on hTCs. In line with this observation, the SN of Ox-DCs did not stimulate the proliferation of hTCs (Figure 3). However, Ox-DCs themselves elevated the proliferative response of hTCs similarly to dTCs. These findings are seemingly controversial at first glance but may indicate that additional mechanisms might be involved in the beneficial effects of a DCs/TC co-culture. For instance, since DCs are famous for their ability to form physical contact with T cells, it is intriguing to suggest that Ox-DCs might provide special signals to TC via cell–cell contact. However, this concept remains to be tested in future studies.

One striking finding of our study was that the SNs of Ox-DCs, as well as recombinant AREG, affected dTCs but not hTCs. Our data demonstrate that this is likely due to an increased expression of the EGFR on dTCs compared to the hTCs. One explanation for the differences in EGFR expression in the two types of TC is that dTCs and hTCs were isolated from different tendons and different donors. Yet, it is important to mention that EGFR is upregulated during inflammation in different cell types and models [54,55,56]. Thu, it is plausible that cells isolated from an inflamed tendon may express higher levels of EGFR than tenocytes from a healthy tissue, and the triggering of AREG via EGFR may contribute to tenocyte growth. Interestingly, mice with a deficient EGFR signaling pathway have not been reported to have problems with tendons or reduced mobility typical for tendinopathic patients. So, it is tempting to speculate that the AREG/EGFR axis may not be essential for the homeostasis of the tendon tissue in healthy people, but rather activated during inflammation to promote tissue repair. Interestingly, skin FBs were not responding to Ox-DCs or AREG, indicating that the AREG/EGFR axis probably does not stimulate these cells but may have inhibitory effects on collagen production.

Taken together, the results of our study demonstrate for the first time a potential role for human DCs and the AREG-EGFR axis in the regulation of tendon integrity and function. It will be interesting to evaluate in further studies how DCs and macrophages cooperate to regulate inflammation in diseased tendons. AREG produced by DCs is one factor that might be involved in this interplay between immune cells and tenocytes, and which is likely to contribute, together with other factors, such as TGF-β, to repair tendinopathies.

## 4. Materials and Methods

### 4.1. Isolation of PBMCs and Generation of Monocyte-Derived Dendritic Cells

Human monocytes were isolated from anonymized Buffy Coats which were bought from the Austrian Red Cross. According to Austrian law and the assessment of the Ethical Committee, the use of anonymized, not identifiable cells obtained from a commercial provider does not require specific ethical approval. This position is based on the Preamble of “wma declaration of helsinki—ethical principles for medical research involving human subjects”.

Peripheral blood mononuclear cells (PBMCs) were isolated via standard density gradient centrifugation using Ficoll-Paque TM Plus (GE Healthcare, Hatfield, UK) and monocytes were purified from PBMCs using the MACS system (Miltenyi Biotec, Bergisch, Germany) as described previously [15]. CD14+ monocytes were cultured for 6 days in Roswell Park Memorial Institute (RPMI) 1640 culture medium (Sigma-Aldrich, St. Louis, MO, USA, cat. Number R0883) supplemented with 0.5% L-glutamine (L-Glu, Gibco by Thermo Fisher Scientific, Paisly, UK, cat. Number 11539876), 1% penicillin/streptomycin (P/S, PAA Laboratories, Pasching, Austria), and 10% heat-inactivated fetal bovine serum (FBS, Gibco by Thermo Fisher Scientific, UK, cat. Number 10270-106). For triggering differentiation to monocyte-derived dendritic cells (Mo-DCs), human recombinant cytokines were used at the following concentrations: 50 ng/mL GM-CSF (Novo Nordisk A/S, Bagsværd, Denmark) plus 200 U/mL IL-4 (Novo Nordisk A/S, Bagsværd, Denmark). For the treatment with OxPAPC (Avanti Polaris Lipids, Alabaster, AL, USA, cat. number 870604P), this was used at a final concentration of 30 µg/mL, which is frequently used to treat cells in vitro [15,18,19,20,21] and the cells were further cultivated for 24 h.

### 4.2. Tenocytes and Fibroblasts

Healthy and tendinopathic human tendons were harvested to isolate tendon stem/progenitor cells (TSPCs) following the protocol previously described [57]. The TSPCs were then cultured, and after passage 2 they are referred to as tenocytes. The culture medium used was α-MEM medium, supplemented with 10% FBS (Gibco Ltd., Paisley, UK), 1% P/S (PAA Laboratories, Pasching, Austria), and 0.5% L-Glu (Gibco Ltd., Paisley, UK). Healthy tenocytes were obtained from unsuitable parts of semitendinosus tissue used for autologous transplantation in patients following the reconstruction of the anterior cruciate ligament. Diseased tenocytes were isolated from patients that underwent surgery due to tendinopathic injuries of the Achilles tendon in the lower leg. All tendon samples were collected after informed consent according to protocols approved by the Institutional Review Board of San Giovanni di Dio e Ruggi D’Aragona Hospital in Salerno, Italy (Review Board prot./SCCE n. 151, achieved on 29 October 2020). The fibroblasts served as a model for tenocytes due to their structural similarity. Human fibroblasts were obtained from skin samples from anonymous healthy female donors during plastic surgery procedures from the abdomen, breast or back. The study was approved by the ethics committee of the Medical University of Vienna (ECS 1969/2021, approved on 30 October 2023) and conducted according to the principles of the Declaration of Helsinki. Written informed consent was obtained from all participants. The isolation was conducted as previously described [58,59] and cells were cultured in T-75 flasks (Corning Costar, Mesa, AZ, USA) in Dulbecco’s Modified Eagle Medium (DMEM) supplemented with 10% FBS, 1% P/S.

### 4.3. Cell Proliferation Assay

Dendritic cells were seeded in 96-well plates (Corning Costar, Mesa, AZ, USA) in co-culture with either tenocytes or fibroblasts for 48 h (5 × 10^3^ per well for each cell type). Cells were labeled with 0.05 mCi/well of [methyl-^3^H]-thymidine (Perkin Elmer, Waltham, MA, USA) and cultured for 18 h at 37 °C, then harvested using a Unifilter-96 Cell Harvester SS (Perkin Elmer, Waltham, MA, USA). The filter plate was dried for 3 h in a dry incubator. Subsequently, the scintillation cocktail (Perkin Elmer, Waltham, MA, USA) was added to each well (25 µL/well) and the thymidine incorporation was measured by a MicroBeta Microplate Counter (Perkin Elmer, Waltham, MA, USA). Readings are displayed as counts per minute (cpm).

### 4.4. Collagen Production Analysis

For the assessment of collagen production, an enzyme-linked immunosorbent assay (ELISA) was performed. Firstly, 5 × 10^3^ cells/well were seeded and incubated for 48 h. Cells were then removed by washing with distilled water. After ensuring that no cells remained, a washing step with washing buffer (PBS—Corning, Corning, NY, USA 21-040-CV and Tween 0.05%, Bio-Rad Laboratories, Vienna, Austria) was performed and the detection antibody for human pro-collagen I α 1 DuoSet ELISA (R&D Systems, Minneapolis, MN, USA) was added as final concentration of 50 ng/mL. After 2 h incubation at RT, a washing step to remove unbound antibodies was performed. Next, streptavidin-HRP (R&D Systems, Minneapolis, MN, USA) was incubated for 30 min, followed by the addition of the TMB-substrate (reagent A and B, 1:1, R&D System). After 30 min of incubation in the dark, the stop solution was added. Finally, the optical density (OD) was measured with Biosan HiPo Microplate Reader MPP-96 (Quant Assay Software 0.8.1.5) set to 450 nm with wavelength correction set at 570 nm.

### 4.5. RNA Isolation, cDNA Preparation, and Real-Time RT-PCR

DCs were stimulated for 6 h with OxPAPC for the measurement of AREG. RNA was isolated using TRI reagent (Sigma-Aldrich) according to the manufacturer’s protocol. One microgram of total RNA was reverse transcribed with MuLV-RT using oligo(dT) primers. PCR primers were designed using the PRIMER3 software 4.1.0, from the Whitehead Institute for Biomedical Research. The amplified cDNA regions were chosen to span one or more large introns in the genomic sequence. The testing of primer specificity included melting point analyses, agarose gel electrophoresis of the PCR products, and subsequent DNA sequencing. PCR was performed as described elsewhere [15,60]. Primer sequences used are as follows: AREG forward 5′-CGG GAG CCG ACT ATG ACTAC-3′; AREG backward 5′-CCA TTT TTG CCT CCC TTT TT-3′. Quantitative RT-PCR was performed using a LightCycler (Roche Molecular Biochemicals, Mannheim, Germany) using SYBR Green I detection.

### 4.6. Determination of AREG Production

DCs were treated as indicated, and after 24 h, the supernatants were harvested and analyzed by ELISA. AREG was measured by sandwich ELISA using Maxisorp plates (Nunc, Roskilde, Denmark) and DuoSet human amphiregulin ELISA (R&D Systems, Minneapolis, MN, USA) according to the instructions of the manufacturer. Assays were performed in duplicates.

### 4.7. Flow Cytometry

For cell surface staining, 5 × 10^4^ cells were incubated with the unconjugated antibodies, anti-hEGFR (Invitrogen, Carlsbad, CA, USA, #608726A, 31G7), for 30 min at 4 °C. Two washing steps were performed. Then, the samples were incubated for another 30 min with the secondary antibody, Alexa Fluor 488 goat anti-mouse IgG (#2379467, 1:100, Invitrogen, Waltham, MA, USA). After further washing steps, flow cytometry data were acquired using a FACS Calibur with CellQuest software 6.0 and LSR Fortessa (both BD Bioscience, Franklin Lakes, NJ, USA).

For the viability assay, tenocytes and fibroblasts (3 × 10^4^), either untreated (mock) or treated with OxPAPC (30 µg/mL), were seeded in 24-well plates. Three different time points (0, 24, and 48 h) were analyzed. At each time point, cells were harvested and collected in FACS tubes (Micronic, lot. #230553/230331/60, Lelystad, The Nerherlands) and washed twice with a staining buffer (1% FBS in PBS 1×). Following the washing steps, 50 µL/mL of 100 ng/mL of propidium iodide (PI, lot. #P-1304, Invitrogen) was added to the tubes. Finally, the data were acquired using a FACS LSR Fortessa.

### 4.8. Statistical Analysis

Statistical analyses were performed using GraphPad Prism 9.0–10.0 Software (GraphPad Software, Inc., La Jolla, CA, USA). For comparisons, *t* tests or one-way ANOVA were used followed by Dunnett’s multiple comparison post hoc test. Data of replicates were entered, and then plotted as mean with SEM, error bars were set automatically. *p*-values < 0.05 were considered as significant. Significant values were marked as one star (*) to indicate *p* < 0.05, two stars (**) to indicate *p* < 0.01, three stars (***) to indicate *p* < 0.001, and four stars (****) to indicate *p* < 0.0001. Flow cytometry data were analyzed using FlowJo™ software version 10.6.1 (Becton Dickinson, Franklin Lakes, NJ, USA).

## Figures and Tables

**Figure 1 ijms-25-07600-f001:**
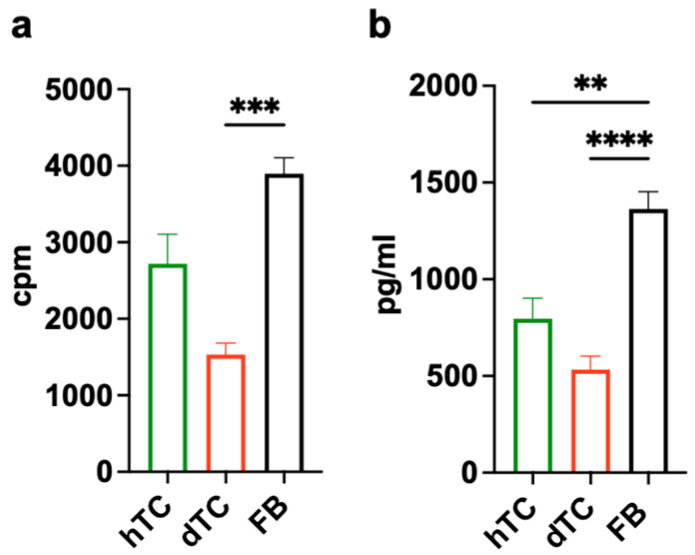
Proliferation rates and collagen production of tenocytes and fibroblasts. (**a**) Proliferation of hTCs (green), dTCs (red), and FBs (black) was measured on day 2 via thymidine incorporation. Data are expressed as counts per minute (cpm). (**b**) Produced collagen by tenocytes and fibroblasts was measured by ELISA after 2 days of culture. Mean ± SEM from 3 independent experiments is shown. A one-way ANOVA test was performed. Only significant differences are indicated. ** = *p* < 0.01; *** = *p* < 0.001; **** = *p* < 0.0001.

**Figure 2 ijms-25-07600-f002:**
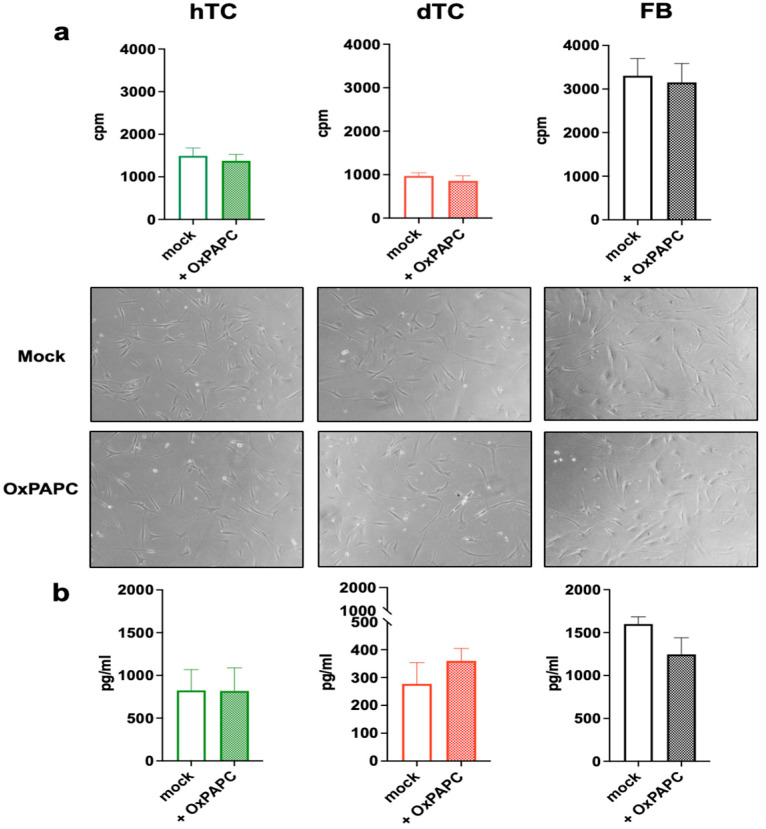
OxPAPC does not influence either proliferation or collagen production of tenocytes and fibroblasts. (**a**) hTCs (green), dTCs (red), and FBs (black) were cultured with or without OxPAPC (30 μg/mL) and proliferation was then measured on day 2 via thymidine incorporation. Data are expressed as counts per minute (cpm). Brightfield images illustrate no morphological changes in TC or FBs after OxPAPC treatment. Scale bar = 300 μm. (**b**) Tenocytes and fibroblasts were cultured for 48 h with or without OxPAPC and the collagen produced was detected by ELISA. Data are expressed as mean ± SEM from 3 independent experiments. A one-way ANOVA test was performed. Only significant differences are indicated.

**Figure 3 ijms-25-07600-f003:**
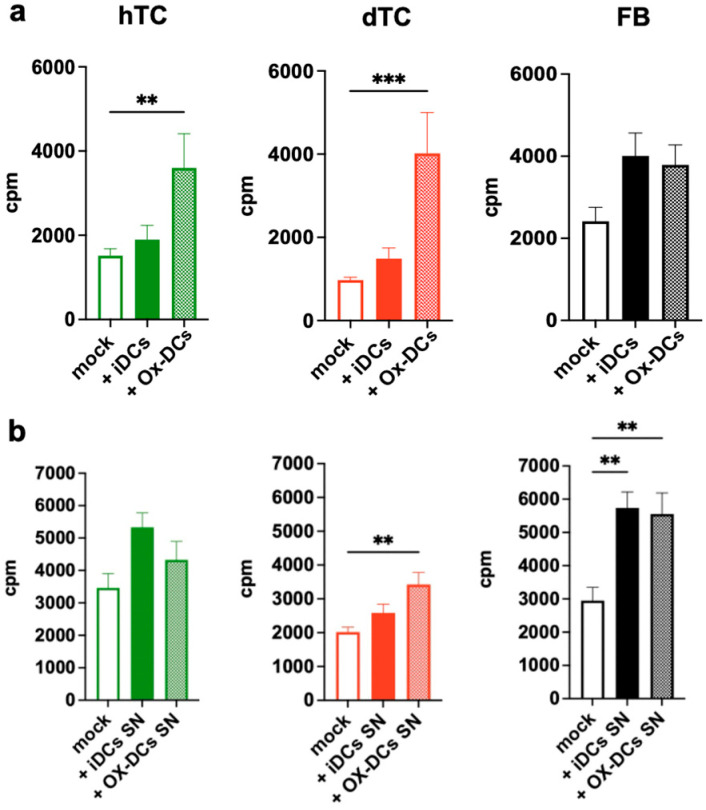
Ox-DCs and their SN are effective in stimulating diseased tenocyte proliferation. (**a**) hTCs (green), dTCs (red), and FBs (black) were co-cultured with (**a**) Ox-DCs or (**b**) treated with Ox-DCs SN and the proliferation was then evaluated on day 2 via thymidine incorporation. Data are expressed as count per minute (cpm). Data are displayed as mean ± SEM from 3 independent experiments. A one-way ANOVA test was performed. Only significant differences are indicated. ** = *p* < 0.01; *** = *p* < 0.001.

**Figure 4 ijms-25-07600-f004:**
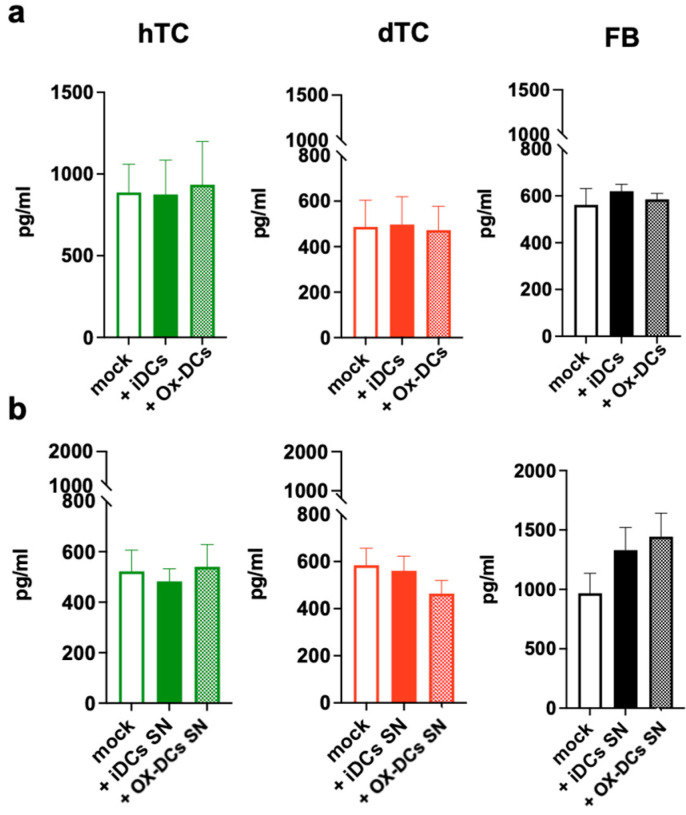
Ox-DCs do not affect collagen production. (**a**) iDCs and Ox-DCs were co-cultured with hTCs (green), dTCs (red), and FBs (black) for 2 days and then the produced collagen was measured via ELISA. (**b**) Tenocytes and fibroblasts were treated with either iDC SNs or Ox-DC SNs for 2 days and the produced collagen was then detected via ELISA. Values are presented as means ± SEM from 3 experiments. Differences between groups were tested using one-way ANOVA. Only significant differences are shown.

**Figure 5 ijms-25-07600-f005:**
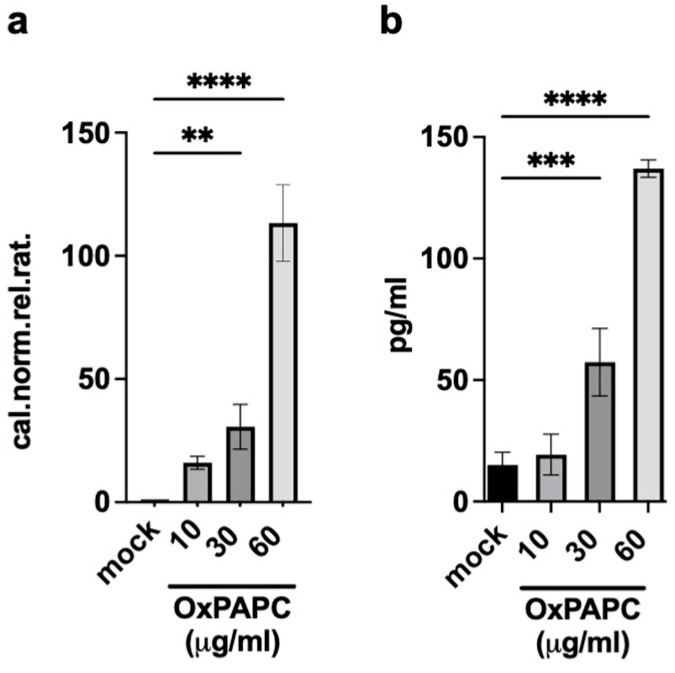
Ox-DCs produce and release amphiregulin. (**a**) Quantitative real-time PCR of monocyte-derived human DC stimulated for 6 h with OxPAPC. A specific primer for human amphiregulin (AREG) was used and results were normalized against GADPH. Mean values from duplicates of a representative experiment are shown. (**b**) AREG release from DC stimulated for 24 h was measured by ELISA. Mean ± SEM from 3 different experiments is shown. A paired Student *t*-test was performed. ** = *p* < 0.01; *** = *p* < 0.001; **** = *p* < 0.0001.

**Figure 6 ijms-25-07600-f006:**
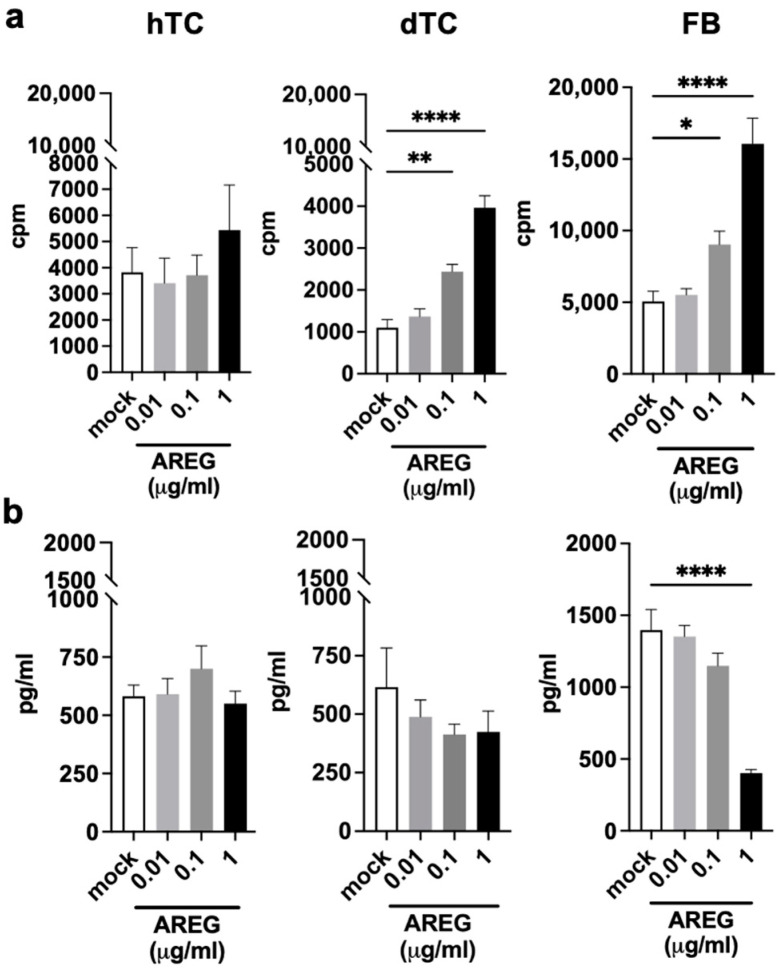
AREG is a growth factor for dTCs and FBs and does not modulate collagen production of tenocytes. (**a**) hTCs, dTCs, and FBs were treated for 2 days with increasing amounts of human recombinant AREG and the proliferation rates were measured by thymidine incorporation. Data are expressed as counts per minute (cpm). (**b**) Tenocyte or fibroblast collagen production was evaluated by ELISA after 2 days of treatment with increasing amounts of AREG. Mean ± SEM from 3 independent experiments is shown. A one-way ANOVA test was performed. Only significant differences are indicated. * = *p* < 0.05; ** = *p* < 0.01; **** = *p* < 0.0001.

**Figure 7 ijms-25-07600-f007:**
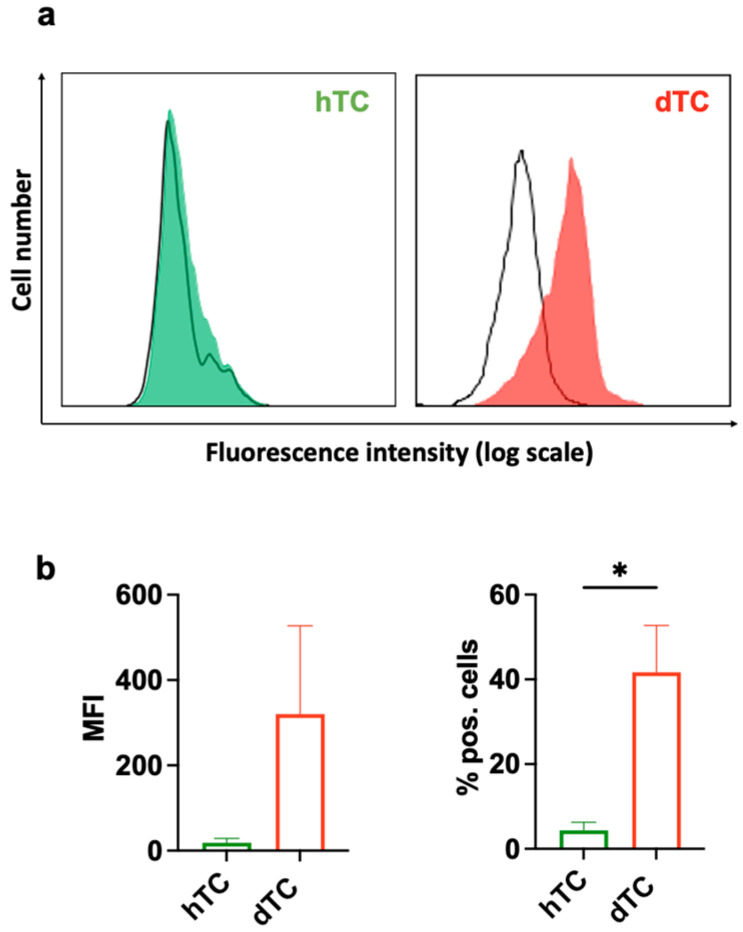
Diseased tenocytes express EGFR. (**a**) EGFR surface expression profile in hTCs (green) and dTCs (red) was measured by flow cytometry. Negative control is shown by the black line histogram. Log scale, logarithmic scale. (**b**) Bar graphs show the quantification of mean fluorescent intensity (MFI) and the percentage of positive cells of hTCs and dTCs. Data are displayed as mean ± SEM from 3 independent experiments. * = *p* < 0.05.

## Data Availability

Data is contained within the article and Appendix A.

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
