# Peer review of "Oxidized Phospholipids Regulate Tenocyte Function via Induction of Amphiregulin in Dendritic Cells"

_ijms, 2024, doi:10.3390/ijms25147600_

Round 1
Reviewer 1 Report
Comments and Suggestions for Authors
In this paper, the authors first investigated the impact of oxidized phospholipids (OxPAPC) on the function of human tenocytes and then analyzed how OxPAPC-treated DCs might affect the fate of tenocytes. The major conclusion is that OxPAPC-treated DCs are able to modulate the proliferation of tenocytes through the AREG-EGFR axis that implies the importance of this specific communication in the repair and regeneration of tendon tissue.
The manuscript is clearly written with no typos. The experiments are well described, the representation of data is clear and easy to follow, and the interpretation of the data is proper. The overall quality of the manuscript is excellent.
After careful review, the authors could consider the following suggestions to improve the quality of the manuscript:
- It should be explained in the text why the 30 µg/ml concentration of OxPAPC was used to treat the cells. For instance, a titration curve could be shown as a supplementary material. If the dose was chosen based on prior studies, it should also be indicated.
- The authors stated that based on microarray data 733 genes were upregulated, and among those, AREG was one of the most prominent. The data of the microarray analysis should be shown in the manuscript (e.g. visualized as a scatter plot, Venn diagram, or volcano plot), and it should also be highlighted which genes were upregulated similarly to AREG. This would highly increase the quality of the current manuscript.
Minor errors:
Lane 70: use “mode of action of OxPLs” instead of “mode of action OxPLs“
Lane 367: “for another 30 minutes” instead of “for others 30 minutes”
Author Response
Thank you very much for your interest in our study and the constructive evaluation and suggestions. To the raised comments and questions, we would like to respond in the following way:
Comment 1: It should be explained in the text why the 30 µg/ml concentration of OxPAPC was used to treat the cells. For instance, a titration curve could be shown as a supplementary material. If the dose was chosen based on prior studies, it should also be indicated.
Response 1: OxPAPC is typically used in in vitro studies between 10-100 µg/ml. A concentration of 30 µg/ml of OxPAPC is a standard concentration which was and is used by other researchers in the field and was also used in our previous studies (Ref. 15, 18, 19, 20, 21) to analyze the effect of OxPLs on various cell types including immune-, endothelial- and vascular smooth muscle cells. This information is now also provided in the text (line 315-316).
Comment 2: The authors stated that based on microarray data 733 genes were upregulated, and among those, AREG was one of the most prominent. The data of the microarray analysis should be shown in the manuscript (e.g. visualized as a scatter plot, Venn diagram, or volcano plot), and it should also be highlighted which genes were upregulated similarly to AREG. This would highly increase the quality of the current manuscript.
Response 2: We now provide the requested information in our new Supplementary Figure 2 in form of a scatter plot and a table illustrating the 10 genes that were most strongly upregulated. AREG was amongst them with a 51-fold induction upon OxPAPC stimulation of DCs. This information is now also provided in the text (line 163).
Comment 3:
Lane 70: use “mode of action of OxPLs” instead of “mode of action OxPLs“
Lane 367: “for another 30 minutes” instead of “for others 30 minutes”
Response 3: Thank you for your comments, we have now changed the wording in both sentences (now Lane 72 and Lane 385).
Reviewer 2 Report
Comments and Suggestions for Authors
In this study, Pinnaro et al. investigate the effect of oxidized phospholipids (OxPAC) on the proliferation and collagen production capabilities of primary tenocytes. The authors used a sufficient spectrum of techniques to analyze the main questions of this topic. However, some additional experiments and some controls are missing (see major points for the text).
Overall the manuscript is understandable and the story line is clear. However, some questions arise for the results obtained from the fibroblast control cells (FB). Moreover, intensive corrections concerning spelling, grammar and phrasing at some points in the text are mandatory.
If the points mentioned in detail below can be addressed by the authors in a major revision, this already fine article is ready for publication and will be a useful contribution to the field.
Major experimental points:
· In Figure 1A, B: Please add experimental data for a starting point of the treatment (t 0h), because this will give a better insight in how quick and how much the proliferation and production capabilities of the cell types changed over the time. It is clear that there are differences at the investigated time point, but it is necessary to compare these findings with the starting levels of both proliferation and collagen production. Please also add significances to the data sets of hTC and dTC.
· The authors state in line 100 – 101 that there was no impact on morphology or viability. The microscopic pictures are sufficient for the morphological part but this is not enough evidence to state that there is no impact on viability. Any kind of cell death measurement (LDH, CellTiter Glo, Trypan blue) is sufficient here to generate data concerning the viability. Please provide these data with and without OxPAC treatment for all three cell types and compare non-treated with treated cells for calculations.
· The authors also state in lines 125-126 that treatment with Ox-DCs has an impact on all three cell types. However, the data clearly show, while this statement is true for dTCs and hTCs, that it does no matter for FB, if they are treated with iDC or Ox-DC. Both DC cell types or their supernatants induce proliferation in FBs in the exact same extend. Interestingly in Figure 6B, the FBs also react differently that the other cells after the AREG treatment. Both findings are not problematic, however, the authors have to correct the lines 125-126 and discuss this clear difference between FB and dTC/hTC in more detail. This is not a flaw in the study, but remarkable, so it has to be described and discussed more intensively.

Author Response
Thank you very much for your interest in our study and the constructive evaluation and criticism. To the raised comments and question we would like to respond in the following way:
Comment 1: intensive corrections concerning spelling, grammar and phrasing at some points in the text are mandatory
Response 1: In order to improve the quality of language of our text, a colleague and native speaker from the USA – Lois Cavanagh - has performed proof-reading and corrections of the manuscript concerning spelling, grammar and phrasing were made throughout the paper. The changes and corrections are highlighted in yellow.
Comment 2: In Figure 1A, B: Please add experimental data for a starting point of the treatment (t 0h), because this will give a better insight in how quick and how much the proliferation and production capabilities of the cell types changed over the time. It is clear that there are differences at the investigated time point, but it is necessary to compare these findings with the starting levels of both proliferation and collagen production. Please also add significances to the data sets of hTC and dTC.
Response 2: The results in Figure 1 do not show a treatment of our cells but demonstrate the spontaneous capacity to proliferate and to produce collagen after 3 days of culture. During the last 24h thymidine is added and since thymidine incorporation needs time, it is not possible to measure proliferation at a starting point with this method.
Yet, we have analyzed the viability of our cells at t0h with or without OxPAPC treatment. This information is now provided in our new Supplementary Figure 1 and in the text (line 107).
All significant values are shown in the figures of our paper. If significances are missing, the differences in the data were not significant like in the case of Figure 1 for hTC vs. dTC.
Comment 3: The authors state in line 100 – 101 that there was no impact on morphology or viability. The microscopic pictures are sufficient for the morphological part but this is not enough evidence to state that there is no impact on viability. Any kind of cell death measurement (LDH, CellTiter Glo, Trypan blue) is sufficient here to generate data concerning the viability. Please provide these data with and without OxPAC treatment for all three cell types and compare non-treated with treated cells for calculations.
Response 3: Thank you very much for this request. We now present in our new Supplementary Figure 1 that staining of cells with propidium iodide and subsequent analyses by flow cytometry that OxPAPC has no toxic effects on the cells used in this study. This is in line with other studies where the effects of OxPAPC on different cell types have been tested without reporting toxic effects at a concentration of 30 µg/ml.
Comment 4: The authors also state in lines 125-126 that treatment with Ox-DCs has an impact on all three cell types. However, the data clearly show, while this statement is true for dTCs and hTCs, that it does no matter for FB, if they are treated with iDC or Ox-DC. Both DC cell types or their supernatants induce proliferation in FBs in the exact same extend. Interestingly in Figure 6B, the FBs also react differently that the other cells after the AREG treatment. Both findings are not problematic, however, the authors have to correct the lines 125-126 and discuss this clear difference between FB and dTC/hTC in more detail. This is not a flaw in the study, but remarkable, so it has to be described and discussed more intensively.
Response 4: You are right that FB behave different compared to TC and that this should be described and discussed in more detail. In order to make this point clearer, we have now changed the text in the Results section (line 128-133) and in the Discussion (line 284-286).
Round 2
Reviewer 2 Report
Comments and Suggestions for Authors
Thank you very much for adressing my points. This manuscript is ready for puplication.